# River runoff in Switzerland in a changing climate – changes in moderate extremes and their seasonality

Regula Muelchi[1], Ole Rössler[1,2], Jan Schwanbeck[1], Rolf Weingartner[1,3], Olivia Martius[1,3]

[1]Institute of Geography and Oeschger Centre for Climate Change Research, University of Bern, Switzerland
[2] now at: German Federal Institute of Hydrology (BfG), Germany
[3]Mobiliar Lab for Natural Risks, University of Bern, Switzerland

*Correspondence to:* Regula Muelchi (regula.muelchi@giub.unibe.ch)

**Abstract.** Future changes in river runoff will impact many sectors such as agriculture, energy production, or ecosystems. Here, we study changes in the seasonality, frequency, and magnitude of moderate low and high flows and their time of emergence. The time of emergence indicates the timing of significant changes in the flow magnitudes. Daily runoff is simulated for 93 Swiss catchments for the period 1981–2099 under Representative Concentration Pathway 8.5 with 20 climate model chains from the most recent transient Swiss climate change scenarios.

In the present climate, annual low flows typically occur in the summer half-year in lower-lying catchments (<1500 m.a.s.l.) and in the winter half-year in Alpine catchments (>1500 m.a.s.l.). By the end of the 21[st] century, annual low flows are projected to occur in late summer and early autumn in most catchments. This indicates that decreasing precipitation and increasing evapotranspiration in summer and autumn exceed the water contributions from other processes such as snow and glacier melt. In lower-lying catchments, the frequency of annual low flows increases, but their magnitude decreases and becomes more severe. In Alpine catchments, annual low flows occur less often and their magnitude increases. The magnitude of seasonal low flows is projected to decrease in the summer half-year in most catchments and to increase in the winter half-year in Alpine catchments. Early time of emergence is found for annual low flows in Alpine catchments in the 21[st] century due to early changes in low flows in the winter half-year. In lower-lying catchments, significant changes in low flows emerge later in the century.

Annual high flows occur today in lower-lying catchments in the winter half-year and in Alpine catchments in the summer half-year. Climate change will change this seasonality mainly in Alpine catchments with a shift towards earlier seasonality in summer due to the reduced contribution of snow and glacier melt in summer. Annual high flows tend to occur more frequent, and their magnitude increases in most catchments except some Alpine catchments. The magnitude of seasonal high flows in most catchments is projected to increase in the winter half-year and to decrease in the summer half-year. However, the climate model agreement on the sign of change in moderate high flows is weak.

# 1 Introduction

Assessments of climate change impacts on hydrology are crucial for future water management and adaptation planning. In particular, annual and seasonal moderate low and high flows are relevant for several reasons. First, even moderate extremes are important for water management planning. Second, very extreme high flows and very extreme low flows are difficult to simulate, because many processes are not fully understood or not yet resolved in hydrological models. Third, hydrological models are calibrated on observed flow conditions and may miss plausible extreme events that have not been experienced. Fourth, climate change projections incorporate large uncertainties regarding small-scale extreme events, particularly for extremes in precipitation, which are potential flood triggers. Therefore, we focus on moderate extremes: events that occur on average once every year or season in today's climate. The larger number of events increases the robustness of the changes estimated.

Low flows have a strong impact on water quality, freshwater ecosystems, and human water use such as power production, drinking water production, irrigation, fisheries, and recreation (IPCC, 2014). Today, long-term water management planning for Switzerland must rely on low-flow assessments from past observations. Because climate change is projected to alter low-flow characteristics, low-flow projections for the 21$^{st}$ century need to be integrated into water management planning. Changes in low-flow indicators in recent decades have been already identified in Europe (Stahl et al., 2010) and in Switzerland (Weingartner and Schwanbeck, 2020). For Switzerland, increasing low-flow magnitudes in winter and decreasing low-flow magnitudes in summer have been observed in snow-driven and rain-driven catchments. Magnitudes of low flows in glaciated catchments have increased in all seasons (Weingartner and Schwanbeck, 2020). Previous studies have assessed climate change impacts on low flows mainly for macroscale catchments and regions. Van Vliet et al. (2013) investigated low-flow changes on a global scale, and other studies focused on European scales (e.g. Feyen and Dankers, 2009; Forzieri et al., 2014; Alderlieste et al, 2014; Papadimitriou et al., 2016; Vidal et al., 2016; Marx et al., 2018). For Switzerland, previous climate impact studies on low flows exist for lower-lying catchments in the Swiss Plateau (Meyer et al., 2011), for large-scale catchments (Bernhard and Zappa, 2012), and for very extreme low-flow regimes (100-year return periods) in aggregated regions (Brunner et al., 2019a). The studies found decreasing low-flow magnitudes in the lower-lying parts of Central Europe but increasing low-flow magnitudes in Alpine areas, where runoff generation is mainly dominated by snow and glacier melt.

High flows can cause damage and costs and are also important for water management and ecology. Assessing future changes in the magnitude, frequency, and timing of high flows is thus crucial for planning and decision making. Previous studies have investigated past trends in floods in Europe (e.g. Stahl et al., 2012; Hall et al, 2014; Mangini et al., 2018; Blöschl et al., 2019; Bertola et al., 2020) and in Switzerland (e.g. Birsan et al., 2005; Allamano et al., 2009; Schmocker-Fackl and Naef, 2010a,b; Castellarin and Pistocchi, 2012). No clear or significant trend in flood magnitude was found by these studies because they disagree on the direction of trends. Various factors make it difficult to compare trends in flood magnitude between catchments and between studies. The assessments depend strongly on the quality and homogeneity of the observations, the underlying methods such as the selection of indicators and statistical tests, and the time periods investigated. Flood frequencies have

increased in northern Switzerland and decreased in southern Switzerland in the recent past (Schmocker-Fackl and Naef, 2010a, Blöschl et al., 2019). Periods with many floods were found in the end of the 19[th] century and after 1968 in northern Switzerland (Schmocker-Fackl and Naef, 2010a). Several studies have also assessed future changes in floods in Switzerland (Allamano et al., 2009; Köplin et al., 2014; Beniston et al., 2016; Ragettli et al., 2020). Although these studies differ substantially in methodological aspects and catchment selection, they found generally increasing but not necessarily significant changes in annual runoff maxima under climate change. Seasonal patterns of change were detected, with increasing flood magnitudes in winter and decreasing flood magnitudes in summer (Allamano et al., 2009). Future shifts in the seasonality of floods also depend on the regime type of the catchments (Köplin et al. 2014).

Here, we complement these assessments with a focus on moderate low and moderate high flows: 7-day runoff minima and daily runoff maxima. The new Hydro-CH2018-Runoff dataset (Muelchi et al., 2020; Muelchi et al., 2021a) is used. It consists of daily runoff simulations for 119 years (1981–2099) driven by the most up-to-date climate change scenarios for Switzerland CH2018 (CH2018, 2018). For the Representative Concentration Pathway RCP8.5 (Moss et al., 2010; van Vuuren et al., 2011), we analyse (1) changes of moderate low and high flows under climate change, (2) the point in time when significant changes emerge, (3) changes in the seasonality of moderate extremes, and (4) changes in their frequency and their co-occurrence. In a companion paper, Muelchi et al. (2021b) assessed changes in runoff regimes and their time of emergence. Here, we extend this analysis with assessments of moderate low and high flows. Because both studies are based on the same simulations (Hydro-CH2018-Runoff ensemble), they complement each other and provide a comprehensive overview of hydrological changes in Switzerland. They also complement the existing studies on future changes in extreme hydrological events.

## 2 Data

We analyse daily runoff simulations for 93 catchments in Switzerland (catchment areas range between 14 km$^2$ and 1700 km$^2$) and covering a wide range of runoff regime types, including glaciated catchments (glaciation between 0.2% and 22%), mainly snow-driven catchments in the Alpine area and lower-lying catchments mainly driven by precipitation and evapotranspiration. Due to strong elevation dependence in the runoff response to climate change (Koeplin et al., 2014, Muelchi et al., 2021b), the catchments are divided into two groups: 39 catchments in the Alpine area with mean altitudes greater than 1500 m.a.s.l. (including 22 glaciated catchments) and 54 catchments in the lower-lying areas in Switzerland with mean altitudes lower than 1500 m.a.s.l. The locations of the catchments are depicted in Fig. 1, with six representative catchments highlighted in blue. These representative catchments cover the most important regime types in Switzerland (Weingartner & Aschwanden, 1992): Rosegbach, highly glaciated (22%); Kander, partially glaciated (5%); Plessur, Alpine snow influenced; Emme, pre-Alpine rain and snow influenced; Venoge, lowland rain dominated; and Verzasca, southern Alpine rain and snow dominated.

The data used for the analysis is the Hydro-CH2018-Runoff ensemble, consisting of daily mean runoff simulations for each of these 93 catchments (Muelchi et al., 2020; Muelchi et al., 2021a). These simulations were run with the semidistributed

PREecipitation-Runoff-EVApotranspiration HRU Model (PREVAH) hydrological modelling system (Viviroli et al., 2009). PREVAH accounts for important hydrological processes such as evapotranspiration, soil moisture dynamics, snow accumulation, and snow melt. A glacier module has been incorporated to account for glacier melt in glaciated catchments. PREVAH was calibrated with even years between 1985 and 2014 and validated with odd years between 1985 and 2014 for each of the 93 catchments individually. Using observed discharge for calibration may overemphasize high-flow conditions and overestimate low-flow conditions. Therefore, the calibration was performed on four groups of observations: daily discharge measurements, transformed daily discharge, monthly mean runoff, and the annual volume. This ensures good performance for the general catchment response to meteorological forcing and for the discharge volume. The hydrological model is driven by daily temperature and precipitation data for each catchment separately from the new high-resolution (2 by 2 km) climate change scenarios for Switzerland CH2018 (CH2018, 2018). In nonglaciated catchments, land use was assumed to be constant over the simulation period. In glaciated catchments, the glaciated area was updated every 5 years in line with glacier projections by Zekollari et al. (2019), which were driven by the same climate model chains. Land use in areas where glaciers disappear during the simulation period were replaced by bare soil for areas below 3000 m.a.s.l. and by rock for areas above 3000 m.a.s.l. The Hydro-CH2018-Runoff ensemble includes simulations for three Representative Concentration Pathways (RCPs): RCP2.6, RCP4.5, and RCP8.5. We constrained our analysis to the RCP8.5 pathway (Moss et al., 2010; van Vuuren et al., 2011) for three reasons. Firstly, the RCP8.5 pathway is considered the worst-case scenario, and simulations based on this RCP pathway are expected to cover the full range of changes likely to occur. Secondly, the changes in low and high flows from these simulations are more pronounced and thus easier to interpret than simulations based on the RCP2.6 and RCP4.5 pathways, although the direction of change in most catchments is the same. Finally, the RCP8.5 pathway includes the largest number of model chains. The larger the number of simulations within an emissions pathway, the more robust are the results, and this is particularly relevant for analysing the time of emergence. In total, 20 daily simulations under the RCP8.5 emission pathway for the period 1981–2099 are available for each of the 93 catchments. Table 1 shows the climate model combinations used in this study.

## 3 Methods

Several indicators for low-flow analysis focus on various properties of low flows (Tallaksen and Van Lanen, 2004). For low flows, we use the minimum 7-day moving average runoff (MAM7) within a half-year or a year. This indicator is proposed by the Swiss Federal Office for the Environment (FOEN) for low-flow statistics. The 30-year average of MAM7 is then considered as moderate low flow and used to assess changes in moderate low flows under climate change. For moderate high flows, we use the 30-year average of the annual maxima per half-year or year as a moderate high-flow indicator. The seasons are defined as summer half-year (May to October) and winter half-year (November to April). The seasonal distinction is used because low and high flows in the winter and summer half-years are governed by different processes and have different impacts.

Percentage changes are calculated for each simulation as the relative difference between the 30-year mean for the future period (2070–2099) and the 30-year mean for the reference period (1981–2010). The multimodel median for 20 simulations of the relative changes by the end of the century is regarded as the best estimate. To indicate the robustness of the projected changes, catchments are highlighted in the figures when at least 90% of the simulations show the same direction of change. This corresponds to "very likely" in the terminology of the Intergovernmental Panel on Climate Change (IPCC) that changes in moderate low and high flows are either positive or negative (Mastrandrea et al., 2010).

To evaluate potential changes in seasonality, the day of the year for each low-flow and high-flow event is extracted. Because moderate low flows are calculated from 7-day averages, the last day of the 7-day period is considered the day of the low-flow event. Median seasonality is then derived by transforming the day of the year into angular values and calculate the circular median. Finally, the angular values are transformed back to the day of the year.

The time of emergence is defined as the time when significant changes in the distribution of moderate low-flow and high flow magnitudes emerge from natural variability (Giorgi and Bi, 2009; Leng et al., 2016). For each simulation and catchment, moderate low-flow and high-flow magnitude distributions of moving 30-year windows are tested against the 30-year reference period using a Kolmogorov-Smirnov test. This test has been found to result in a more robust and earlier estimation of time of emergence than other methods (Gaetani et al., 2020). The time of emergence is then defined following Mahlstein et al. (2011) as the last year of the first 30-year moving window where the Kolmogorov-Smirnov test is rejected with a $p$-value lower than 0.05 (95% significance). We consider the time of emergence robust when at least 66% of the models detect a significant change in the same 30-year window for the first time. This threshold was also used by Mahlstein et al. (2011) and is referred to as "likely" in the IPCC terminology (Mastrandrea et al., 2010). The testing procedure described above also has some disadvantages. Because runoff responses to climate change are subject to natural climate variability, runoff responses may not always show significant changes after the first detection of a time of emergence, so changes in moderate low and high flows may not be significant in all subsequent periods. Therefore, we also analyse the behaviour of the $p$-values over time (see supplement of this paper) to gauge the persistence of the significance of changes.

Changes in the frequency of moderate low and high flows are quantified by counting years when a predefined runoff threshold is exceeded or undercut. We use the median magnitudes of moderate low and high flows in the reference period as thresholds. For moderate high flows, we count years with high flows exceeding this threshold. For low flows, we consider years with low flows below the threshold. This is calculated separately for each seasonal and annual time window and each simulation. Finally, the percentual change in frequency is calculated. We also investigate the co-occurrence of moderate low and high flows. Co-occurrence is defined as high flows exceeding the reference threshold and low flows undercutting the reference threshold in the same year, winter half-year, or summer half-year.

# 4 Results

## 4.1 Future changes in moderate low flows

Median seasonality of moderate annual low flows is shown in Fig. 2 (left panels) for the reference period and by the end of the century. In most Alpine catchments, annual low flows occur in late winter or early spring in the reference period. By the end of the century, low flows occur in autumn. However, low flows in very high Alpine catchments do not change their seasonality. Median seasonality of low flows in pre-Alpine catchments shifts from late autumn to early autumn. In southern Alpine catchments, the median seasonality of low flows changes from winter and spring to early autumn. No clear change in seasonality is found for lower-lying catchments with low flows, which occur in late summer and early autumn. Except in very high Alpine catchments low flows occur between August and October by the end of the century.

The annual low-flow magnitudes show distinctly different patterns of change for Alpine and non-Alpine catchments (Fig. 3 left panels). The scale bar is limited to -60% and +60% for readability. Whereas the magnitude of annual low flows decreases by up to -66% in most of the lower-lying catchments, the Alpine catchments show strong increases in magnitude (up to +200%). Low flows in the winter half-year in Alpine catchments coincide with the typical low-flow season in the reference period, but low flows in the summer half-year coincide with the typical low-flow season in lower-lying catchments. Low-flow magnitudes in the winter half-year increase on average by +22%. Increases are found in two thirds of the catchments, again with stronger increases in very high Alpine catchments. In the summer half-year, the low-flow magnitudes decrease on average by -40% (maximum decrease -74%). However, three high Alpine catchments still show an increase in low-flow magnitudes in the summer half-year due to an increase in low flows in May. More catchments show good model agreement (>90%) in the summer half-year (87 catchments) than in the annual (63) low flows and in the winter half-year (30).

Transient changes of moderate low-flow magnitudes and seasonality throughout the 21[st] century are shown for three representative Alpine catchments and three representative lower-lying catchments in Figs. 4 and 5, respectively. The relative changes for each of the catchments and each of the time windows are summarized in Table 2. The high Alpine catchment, Rosegbach, shows strong increases in both annual low-flow magnitudes and low-flow magnitudes in the two half-year periods (Fig. 4 top row). Although the seasonality of the annual low flows and low flows in the winter half-year does not change, the seasonality of low flows in the summer half-year shifts from early summer to autumn. This indicates a change in the underlying processes, leading to low flows in the summer half-year: The catchment is highly glaciated in the reference period, with a glacier coverage of 22%, but loses most of the glacier coverage by the end of the century (glacier coverage: 1%). In the reference period, the retention of water in snow and ice still takes place in May. Under climate change, enhanced snowmelt increases runoff in early summer. Consequently, the contribution of snow and glacier melt in summer decreases. At the same time, precipitation in summer also decreases. This leads to a strong decrease in summer runoff. The combination of increasing runoff in early summer and decreasing runoff in late summer and early autumn results in a shift of low-flow seasonality in the summer half-year from early summer to early autumn. In the Kander catchment, which has little glacier influence (Fig. 4 middle row), magnitudes of the annual low flows and low flows in the winter half-year increase mainly due to increased winter

precipitation falling as rain instead of snow. Low-flow magnitudes in the summer half-year decrease by the end of the century. The annual low flows and low flows in the half-year periods occur earlier by the end of the century. The snow influenced catchment, Plessur, shows a strong shift in seasonality in the annual low flows from winter to autumn but no change in the magnitude of annual low flows (Fig. 4 bottom row). Low-flow magnitudes in the winter half-year increase and low flows occur earlier in the season. The magnitudes of low flows in the summer half-year decrease but seasonality does not change. The magnitude of annual low flows and low flows in the half-years decrease by the end of the century in the pre-Alpine snow and rain dominated catchment, Emme. The annual low flows show a clear shift in seasonality from late autumn and early winter to early autumn (Fig. 5 top row). A shift towards earlier seasonality is also found for low flows in the winter half-year but not for low flows in the summer half-year. In the mainly rain- and evaporation-driven catchment, Venoge, the annual low flows and low flows in the summer half-year do not change their seasonality, but low flows in the winter half-year tend to occur earlier in the season (Fig. 5 middle row). The magnitude of annual low flows and low flows in the two half-years decrease. The southern Alpine catchment, Verzasca, shows a decrease in magnitude and a strong shift in seasonality from late winter to early autumn for annual low flows (Fig. 5 bottom row). Low-flow magnitude in the winter half-year increase and low-flow magnitude in the summer half-year decreases, both without change in seasonality.

Fig. 6 (left panels) shows the time of emergence for moderate low flows and low flows in the two half-year periods; this is when at least 66% of the models show significant changes in the distribution. In total, 43 catchments show a time of emergence for annual low flows, with particularly early significant changes in glaciated and/or high Alpine catchments (earliest 2018–2047). A total of 20 catchments shows a time of emergence in low flows in the winter half-year, and these have a mean altitude higher than 1600 m.a.s.l. In the summer half-year, 80 catchments show significant changes in low flows, with an early time of emergence again found in high Alpine catchments but a later time of emergence in lower-lying catchments. Most of the catchments show persistent significant changes after the first detection of a time of emergence (Fig. S1 in the supplement).

## 4.2 Future changes in moderate high flows

The median seasonality of annual high flows is shown in Fig. 2 for the reference period and by the end of the century. In Alpine catchments, the median seasonality shifts from summer to late spring and early summer. However, highly glaciated catchments do not change their high-flow seasonality. Moderate high flows in pre-Alpine catchments occur in spring in the reference period and in winter by the end of the century. A change in seasonality is also found in southern Alpine catchments, where high-low seasonality shifts from late summer and early autumn to late autumn in future. In lower-lying catchments, no change is found in high-flow seasonality.

Relative changes of magnitude for high flows by the end of the century are depicted in Fig. 3 (right panels). The magnitudes of annual high flows increase in 71 catchments (up to +28%) and decrease in 22 catchments (up to -22%). Compared to the changes in low flows, the changes in high-flow magnitudes are smaller. There are no clear spatial patterns or elevation dependences, and good model agreement (>90%) is only found in 12 catchments. High flows in the winter half-year in lower-

lying catchments coincide with the typical high-flow season in the reference period, whereas high flows in Alpine catchments are mainly found in the summer half-year. High-flow magnitudes in the winter half-year increase in all catchments, and model agreement is higher, with 45 of 93 catchments showing good agreement. The strongest increases in magnitude and good model agreement are found in high Alpine catchments. However, high flows in the winter half-year in high Alpine catchments are still small in magnitude. High flow magnitudes in the winter half-year in lower-lying catchments increase only moderately, and model agreement is generally weak. High-flow magnitudes in the summer half-year decrease in 74 catchments (up to -26%) and increase in 19 catchments (up to +15%). The strongest reductions in high-flow magnitudes in the summer half-year are found in high Alpine catchments, including the six catchments showing good model agreement. A spatial cluster of increasing high-flow magnitudes in the summer half-year is found in catchments in north-western Switzerland.

The magnitude of annual high flows and of high flows in the summer half-year in the Rosegbach catchment decrease towards the end of the century and tend to occur earlier in summer, but high-flow magnitudes in the winter half-year increase and tend to occur later in the season (Fig. 7 top row). A similar pattern is also found for the Plessur (Fig. 7 bottom row). In the Kander, the annual high-flow magnitudes increase slightly, shift to earlier in the year, and can also occur in the winter half-year by the end of the century (Fig. 7 middle row). High-flow magnitudes in the winter half-year in the Kander also increase, and high-flow magnitudes in the summer half-year show a small decrease without a significant shift in seasonality. The high flows in the Emme and the Verzasca do not change their seasonality, but high-flow magnitudes in the winter half-year increase and decrease in the summer half-year (Fig. 8 top and bottom rows). The rain driven catchment, Venoge, shows increasing high-flow magnitudes with no change in seasonality (Fig. 8 middle rows).

The time of emergence of moderate high flows is depicted in Fig. 6 (right panels). Compared to moderate low flows, fewer catchments exhibit significant changes, and these catchments are mostly high Alpine catchments. For annual high flows, three high Alpine (>2000 m.a.s.l.) catchments show a time of emergence, with of the earliest in 2078 (2049–2078). The 27 catchments showing significant changes in high flows in the winter half-year (earliest 2044, 2013–2044) are also located in the Alpine ridge (>1500 m.a.s.l. mean altitude). In the summer half-year, only six catchments (>1800 m.a.s.l.) show a time of emergence, and the earliest is 2071 (2042–2071). Most of the few catchments showing a time of emergence also show persistent significant changes for the rest of the century a few years after the first detection (Fig. S2 in the supplement). If the time of emergence analysis revealed a significant deviation from current high- and low-flow magnitudes in a catchment, then this change was most often persistent over time. However, the number of catchments showing a time of emergence was found to be small.

## 4.3 Changes in frequency and co-occurrence of low- and high-flow events

So far, we have assessed changes in the magnitude and seasonality of low and high flows. In this section, we address changes in the frequency and co-occurrence of low-flow and high-flow events. To do this, we need to set a threshold to identify events. The threshold runoff value is defined as the median in the reference period.

Fig. 9 illustrates relative changes in the frequency of low flows (upper panels) and high flows (middle panels) by the end of the century (2070-2099) that fall below or exceed median values. Co-occurrence of low and high flows are defined as the occurrence of a high-flow event and a low-flow event in the same time window (lower panels). The frequency of annual low-flow events increases in 70 catchments. These catchments are mainly found in rain driven catchments and to a lesser extent also in snow driven catchments. Catchments showing less frequent low flows are only found in high Alpine areas. Good model agreement is found in 56 catchments. Low flows also occur more frequent in summer, sometimes occurring every year, in almost all regions except few very high-elevation catchments. Most of the catchments (82) show a good model agreement on the increase of low-flow frequency in the summer half-year. Low flows in the winter half-year tend to occur less often in mountainous areas, but lower-lying catchments still show an increase in frequency. However, only mountainous catchments show good model agreement.

Changes in the frequency of high flows are less clear than for low flows. For annual high flows, 58 catchments show increasing frequency, and 30 catchments show decreasing frequency. However, the changes are often small. No clear spatial or elevation pattern emerges, and model agreement is weak. All catchments will face more years with more frequent high-flow events in the winter half-year than they do today, particularly in the high Alpine regions. In contrast, the frequency of high-flow events in the summer half-year will decrease by the end of the century in most catchments. Model agreement is weaker in the summer half-year than in the winter half-year.

Annual co-occurrence increases in most catchments, particularly in the lower-lying catchments. In high Alpine catchments, this co-occurrence decreases, mainly due to the strong increase in winter runoff. Co-occurrence in the winter half-year decreases mainly in high altitude catchments but also in a few of the lower-lying catchments. In the summer half-year, most catchments (85 catchments) show increasing co-occurrence. Only eight high Alpine catchments show decreasing co-occurrence. In contrast to high-flow frequency, the model agreement for co-occurrence is stronger in the summer half-year (48 catchments) than in the winter half-year (14 catchments).

# 5 Discussion

## 5.1 Changes in moderate low flows

Low flows in Alpine regions typically occur in winter and early spring when precipitation falls as snow and accumulates. Storage as snow limits the direct runoff, and only baseflow runoff occurs in winter. Because higher temperatures result in both more precipitation falling as rain instead of snow and earlier snow melt, the low-flow magnitudes in the winter half-year are projected to increase. Furthermore, seasonality shifts from late winter to late autumn. This shift indicates that snow storage no longer dominates low flows. Instead summer and autumn droughts in combination with lack of snow and glacier melt generally become the main driver of low flows. However, this is not the case in highly glaciated catchments with very high mean altitudes (>2300 m.a.s.l.), where the seasonality of low flows does not change. In the summer half-year, the magnitudes of low flows in

Alpine catchments decrease due to the combination of decreasing summer precipitation, enhanced evapotranspiration, and the reduced contribution of snow and glacier melt to the runoff. Exceptions are the catchments at very high altitudes, which show increasing low-flow magnitudes in the summer half-year by the end of the century. Increasing low-flow magnitudes in the winter half-year in Alpine areas have been identified in observations (Weingartner and Schwanbeck, 2020), and our results show that this trend will continue and intensify with climate change. The findings are also in agreement with results for projections of very extreme low-flow regimes (100-year return period; Brunner et al., 2019a).

In the present climate, low flows occur mostly in late summer and autumn in lower-lying catchments. In these catchments, runoff volumes during low-flow conditions are projected to decrease in all time periods, with the reduction in the summer half-year being much stronger than in the winter half-year. The reasons for the reduction in the summer half-year are decreasing summer precipitation and higher temperatures, which enhance evapotranspiration. The projected low-flow reduction in the summer half-year is in line with observed trends (Weingartner and Schwanbeck, 2020), but the changes are amplified under climate change. Even though the climate change scenarios project increasing winter precipitation, the magnitudes of low flows in the winter half-year are projected to decrease, mainly due to a shift in seasonality from winter to late autumn. The seasonality of annual low flows does not change in mainly rain driven catchments. In pre-Alpine regions, the seasonality of annual low flows shifts from late autumn to early autumn. The snow and rain influenced southern Alpine regions typically undergo two periods of low flows: one in late summer and one in winter, with the winter minimum often being lower in the reference period. Under climate change, the seasonality of low flows shifts from winter to late summer and early autumn. At the same time, runoff in low-flow situations decreases by the end of the century.

The projected changes in low flows imply potential impacts for various sectors. Changes in low flows in the winter half-year have different impacts than those in the summer half-year. The increase in mean runoff (Muelchi et al., 2021b) and in low flows in the winter half-year in Alpine regions may be beneficial for hydropower production, which is particularly sensitive to climate change (Schaefli et al., 2007; Zierl and Bugmann, 2005). Electricity demand in Switzerland is highest in winter (Hakala et al., 2020). Even though heating degree days are projected to decrease in Switzerland (Tschurr et al., 2020), heating demand will still occur under climate change. The projected increase in winter mean runoff and low flows may allow enhanced hydropower production and possibly extend the season of hydropower production. Therefore, changes in mean runoff and low flows due to climate change should be considered in negotiations of future water use concessions (Hakala et al., 2020). However, energy demand may increase in summer due to a projected increase in cooling degree days due to higher summer temperatures (Tschurr et al., 2020). At the same time, water demand for irrigation may increase due to projected prolonged and intensified summer droughts and increased evapotranspiration (CH2018, 2018). In addition, river runoff is needed for cooling infrastructure. Because mean flows (Muelchi et al., 2021b) and low flows in the summer half-year are projected to decrease, this combination may lead to conflicts in water use. Policy makers and water management systems need to adapt to these changes in water availability, and a coordinated water use policy should be introduced to cover potential water shortages in the summer half-year. In recent years, the regional authorities in Switzerland that control public water have reviewed their laws and recommendations regarding minimum residual flows. Under climate change, these minimum residual flows are

projected to decrease in summer in lower-lying catchments, indicating that residual flows defined under current conditions may not suffice to serve important water-related services. One of the measures proposed by the authorities is to increase the residual flow for environmental purposes such as protecting flora, fauna, and sediment transports (Hakala et al., 2020). In Alpine regions, an increase in the residual flow regulations may create potential deficits for hydropower providers. This may reinforce potential conflicts in water use during the summer half-year. The combination of a decrease in low flows in the summer half-year and increasing river water temperatures (Michel et al., 2021) increases water stress for river ecosystems, and some fish species such as trout may lose their habitats (FOEN, 2021).

## 5.2 Changes in moderate high flows

In Alpine areas, magnitudes of annual high flow and high flows in the summer half-year are likely to decrease. This can be explained by the decreasing contribution of melt water together with decreasing summer precipitation and increased evapotranspiration. By the end of the century, Alpine areas will face about half of the present mean runoff in summer (Muelchi et al., 2021b), a decrease that is also reflected in high flows. This is in contrast to the magnitudes of high flows in the winter half-year, which are projected to increase with climate change. However, high-flow magnitudes in the winter half-year are still smaller than high-flow magnitudes in the summer half-year. Decreases in the runoff volume during high flows in the summer half-year and increases in high flows in the winter half-year were also found for mountainous regions by Allamano et al. (2009). However, the decreasing annual high-flow magnitudes in Alpine areas contradict Köplin et al.'s (2014) findings of increasing high-flow magnitudes in the Alpine area. The reason for the difference between results is not entirely clear, but Köplin et al. (2014) consider very extreme floods, whereas this study considers moderate high flows.

Annual high flows in the Alpine region usually occur in the summer half-year, when the snow line is high, melting is in progress, and precipitation intensities are highest. In glaciated catchments, high flows currently occur at the end of summer when glacier melt reaches its peak. In snow driven catchments today, the high flows tend to occur in early summer during the snowmelt. In both regime types, seasonality shifts to earlier months by the end of the century so that the high flows occur earlier in summer. Exceptions are the highly glaciated catchments with high mean elevation, which will also have snow and glacier influence in summer in the future. In these catchments, seasonality hardly changes. Köplin et al. (2014) also found shifts in the seasonality of extreme floods in Alpine areas. Their results show a shift in snow driven catchments from summer to autumn, whereas our results show a shift to earlier spring and early summer.

In lower-lying areas, magnitudes of annual high flows and high flows in the winter half-year tend to increase, although the increase is often not robust across models. In the summer half-year, the high-flow magnitudes tend to decrease again with no robust signals across models. Moderate high flows occur in the winter half-year in rain driven catchments, and this will not change. In catchments partly influenced by snow, where high flows occur in spring, the seasonality shifts from spring to late winter. This shift is in agreement with Köplin et al.'s (2014) results. In the southern Alpine areas, the annual high flows also

tend to increase and shift from late summer and early autumn to late autumn, a change also found by Köplin et al. (2014) for extreme floods.

The increase in moderate high flows in the winter half-year may also be beneficial for energy production because today's hydropower production capacity is limited in winter by the snow storage capacity within Alpine catchments. Under climate change, the increasing moderate high flows in Alpine catchments may create opportunities to help meet energy demand in the winter half-year. Reservoirs may be emptied more slowly due to the increase in mean flows (Muelchi et al., 2021b) and moderate high flows. In summer, the moderate high flows in Alpine catchments are projected to decrease. In lower-lying catchments, where the high-flow season is in the winter half-year, increasing moderate high flows may also indicate an increased risk of winter floods. In contrast to the results for moderate low flows, the climate model agreement on the sign of change is weaker, and thus interpretation of our results becomes more difficult.

## 5.3 Time of emergence of significant changes

Significant and robust changes in the magnitude of moderate low flows emerge mainly for annual low flows and low flows in the summer half-year. The majority of the catchments show a significant change in the magnitude of low flows in the summer half-year. High Alpine catchments show earlier significant changes in low flows in the summer half-year than do lower-lying catchments. Early times of emergence in high Alpine catchments were also found for summer mean flow in Muelchi et al. (2021b). In the winter half-year, only Alpine catchments show a significant change in low flows. The main reasons for this are snowpack-related processes such as the change in precipitation from snow to rain, smaller snow accumulations, and consequent increased direct runoff.

The magnitude of high flows significantly changes for only a few catchments. This is due to the large variability across the climate models. To detect a time of emergence, we require at least 66% of the models to agree on significant changes in the distribution of high flows.

## 5.4 Changes in the frequency and co-occurrence of low- and high-flow events

The frequency of annual moderate low-flow events increases in lower-lying catchments, but fewer low-flow events are detected in Alpine catchments. However, the frequency of low flows in the summer half-year will increase in almost all catchments. In some catchments, this frequency almost doubles. This may have implications in agricultural areas where irrigation plays an important role. High-flow events in the winter half-year will occur more often, but high-flow events in the summer half-year will occur less often. A clear pattern in frequency of annual high-flow events cannot be shown because model agreement is weak. However, most catchments show a tendency towards higher frequency. Co-occurrence of low-flow and high-flow events in the same year increases in most lower-lying catchments. In contrast, high-elevation catchments show decreasing co-occurrence, mainly due to the increase in low flows. The changes in co-occurrence are dominated by changes in low-flow

occurrence. Low flows in lower-lying catchments tend to occur much more often, so co-occurrence also decreases, but the opposite is true for low flows in high Alpine catchments. Co-occurrence of low-flow and high-flow events in the same half-year is important for ecosystems because this may shorten their recovery times. Information about co-occurrence is also important for insurance companies' risk assessments.

## 5.5 Uncertainties and Limitations

Several sources of uncertainties affect the results of this study. A detailed discussion of sources of uncertainty in the hydrological simulations is provided in Muelchi et al. (2021a), and they are summarized only briefly here. Uncertainties arise during each modelling step: with the emission scenario (RCP8.5), the selection of climate models and their boundary conditions, the postprocessing method (Gutiérrez et al., 2018), the hydrological model (Addor et al., 2014) and its calibration, and the underlying glacier projections. All these need to be considered in adaptation planning (e.g., Wilby and Dessai, 2010). A comparison with three versions of hydrological models for three catchments showed good agreement on the direction of change of the hydrological response to climate change (Muelchi, 2021). Therefore, we regard the simulations underlying this study as robust for climate change assessments.

Our results show that projections of moderate high flows are less robust among climate models than are those of moderate low flows. Differences in the projections of moderate high flows arise for several reasons. First, high flows are difficult to model because many different processes interact with each other. In particular, small-scale precipitation patterns have a strong influence on high flows, and the input data from the climate models does not reflect small-scale precipitation processes well (Ban et al., 2015). Second, the uncertainty arising from internal variability of extreme precipitation is large, and this is also reflected in our results. Third, our results represent 30-year averages as well as averages across models. Therefore, a great deal of information is averaged out. Despite these limitations, our results show that it is crucial to take projected changes in moderate runoff extremes into account due to their manifold implications for various sectors, as discussed in sections 5.1 for low flows and 5.2 for high flows.

## 6 Summary and conclusions

This study assesses changes in moderate low and high flows under climate change for 93 catchments in Switzerland. Runoff simulations were driven by the most recent transient climate change scenarios (CH2018) for Switzerland for 1981–2099 based on the RCP8.5 scenario. This study analyses changes not only in the magnitude of these moderate low and high flows but also in their seasonality and frequency. Thanks to the transient property of the simulations, the time of emergence could also be assessed.

The projections indicate that changes in low flows over time depend strongly on elevation. Whereas low-flow magnitudes decrease in lower-lying catchments, they increase in Alpine catchments, thus amplifying observed trends (Weingartner and Schwanbeck, 2020). The results for low-flow magnitudes are in line with the projections of previous studies (e.g. Meyer et al., 2011; Bernhard and Zappa, 2012; Brunner et al., 2019a). Moreover, a shift in seasonality was found, with low flows occurring predominantly in late summer and autumn by the end of the 21st century. Increasing low flows in Alpine regions in the winter half-year may be beneficial for hydropower production. However, decreasing low flows in the summer half-year may induce water use conflicts, especially because water demand in summer may increase in various sectors such as irrigation for agriculture and cooling infrastructures (Brunner et al., 2019b). The pronounced projected decrease in low flows in the summer half-year in almost all the catchments except some high Alpine ones may become one of the most important challenges for water management.

Relative changes in magnitude are smaller for moderate high flows than for low flows. Most of the catchments show an increase in moderate high-flow magnitudes, but the model agreement on the changes is not robust except in a few catchments in northern and north-western Switzerland. High Alpine catchments show a decrease in high-flow magnitudes in the summer half-year, mainly due to reduced melt water, and an increase in high flows in the winter half-year. The magnitude of high flows in the winter half-year in Alpine catchments is much smaller than for those in the summer half-year. Thus, the increasing high-flow magnitudes in Alpine catchments in the winter half-year are not that important hydrologically but may become relevant for ecosystems and energy production. Projected changes in magnitude and shifts in seasonality of moderate high flows in lower-lying catchments are in line with previous studies (e.g. Koeplin et al., 2014; Brunner et al., 2019a). For Alpine catchments, our results do not agree with other projections in magnitude or in some cases in seasonality. This lack of agreement may arise from the various indicators considered. Our study focuses on moderate high flows, but comparative studies focused on extreme high flows, which can be governed by different processes than moderate high flows.

Significant changes in the magnitude of low flows emerge early in the 21st century for high Alpine catchments because of an increase in winter flows. For many lower-lying catchments, a significant decrease in low-flow magnitude in the summer half-year is detected but only later in the 21st century. Changes in the magnitude of high flows are mostly not robust across climate models and thus not significant.

Low-flow events will occur more often in lower-lying catchments and less often in high Alpine catchments. Like the weak signal in the magnitude of high flows, changes in the frequency of high-flow events are also small. However, most of the catchments will experience increasingly frequent high-flow events. An elevational pattern was found for the co-occurrence of moderate low and high-flow events, with increasing co-occurrence in lower-lying catchments and decreasing co-occurrence in high Alpine catchments. This pattern is dominated by changes in the frequency and magnitude of moderate low flows.

**Data availability**

The data used in this study is available under https://doi.org/10.5281/zenodo.3937485 (Muelchi et al., 2020).

**Author contributions**

RM performed the analysis of the results and drafted the manuscript. JS, OR, RW, and OM helped in the interpretation of the results. All authors reviewed the resulting data and assisted with paper writing.

**Competing interests**

The authors declare that they have no conflict of interest.

**Acknowledgements**

Authors would like to thank Harry Zekollari for processing and providing the glacier projections used in this study. We also thank MeteoSwiss and FOEN for providing the data necessary for this study. We acknowledge the funding of the Swiss Federal Office for the Environment under the project Hydro-CH2018.

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

**Table 1: Overview of the climate model chains available and their initial grid spacings of 12 km (EUR-11) and 50 km (EUR-44).**

| Global Climate Model | Regional Climate Model | EUR-11 | EUR-44 |
|---|---|---|---|
| ICHEC-EC-EARTH | KNMI-RACMO22E | | X |
| | DMI-HIRMAM5 | X | |
| | CLMcom-CCLM4-8-17 | X | |
| | CLMcom-CCLM5-0-6 | | X |
| | SMHI-RCA4 | X | |
| MOHC-HadGEM2-ES | CLMcom-CCLM4-8-17 | X | |
| | CLMcom-CCLM5-0-6 | | X |
| | KNMI-RACMO22E | | X |
| | SMHI-RCA4 | X | |
| MPI-M-MPI-ESM-LR | CLMcom-CCLM4-8-17 | X | |
| | CLMcom-CCLM5-0-6 | | X |
| | SMHI-RCA4 | X | |
| | MPI-CSC-REMO2009-2 | X | |
| MIROC-MIROC5 | CLMcom-CCLM5-0-6 | | X |
| | SMHI-RCA4 | | X |
| CCCma-CanESM2 | SMHI-RCA4 | | X |
| CSIRO-QCCCE-CSIRO-Mk3-6-0 | SMHI-RCA4 | | X |
| IPSL-IPSL-CM5A-MR | SMHI-RCA4 | X | |
| NCC-NorESM1-M | SMHI-RCA4 | | X |
| NOAA-GFDL-GFDL-ESM2M | SMHI-RCA4 | | X |

**Table 2: Relative changes (in %) by the end of the century and the seasonality (SEAS) of moderate low and high flows for the six representative catchments. The seasonality indicates the season in which the moderate low and high flows generally occur. Abbreviations: Seasonality in the winter half-year in the reference period and predominantly in the summer half-year in the future period (WS), seasonality in the winter half-year in both periods (WW), seasonality in the summer half-year in the reference period and in the winter half-year in the future period (SW), and seasonality in the summer half-year in both periods (SS).**

| Catchment | Moderate low flows | | | | Moderate high flows | | | |
|-----------|--------|--------|--------|------|--------|--------|--------|------|
|           | ANNUAL | WINTER | SUMMER | SEAS | ANNUAL | WINTER | SUMMER | SEAS |
| **Rosegbach** | +191% | +199% | +89% | WW | -20% | +97% | -25% | SS |
| **Kander** | +20% | +41% | -37% | WW | +8% | +50% | -5% | SS |
| **Plessur** | -1% | +32% | -43% | WS | -10% | +43% | -14% | SS |
| **Emme** | -53% | -17% | -66% | WS | +2% | +13% | -7% | SW |
| **Venoge** | -45% | -16% | -47% | SS | +22% | +25% | +4% | WW |
| **Verzasca** | -22% | +42% | -53% | WS | +6% | +24% | -3% | SS |

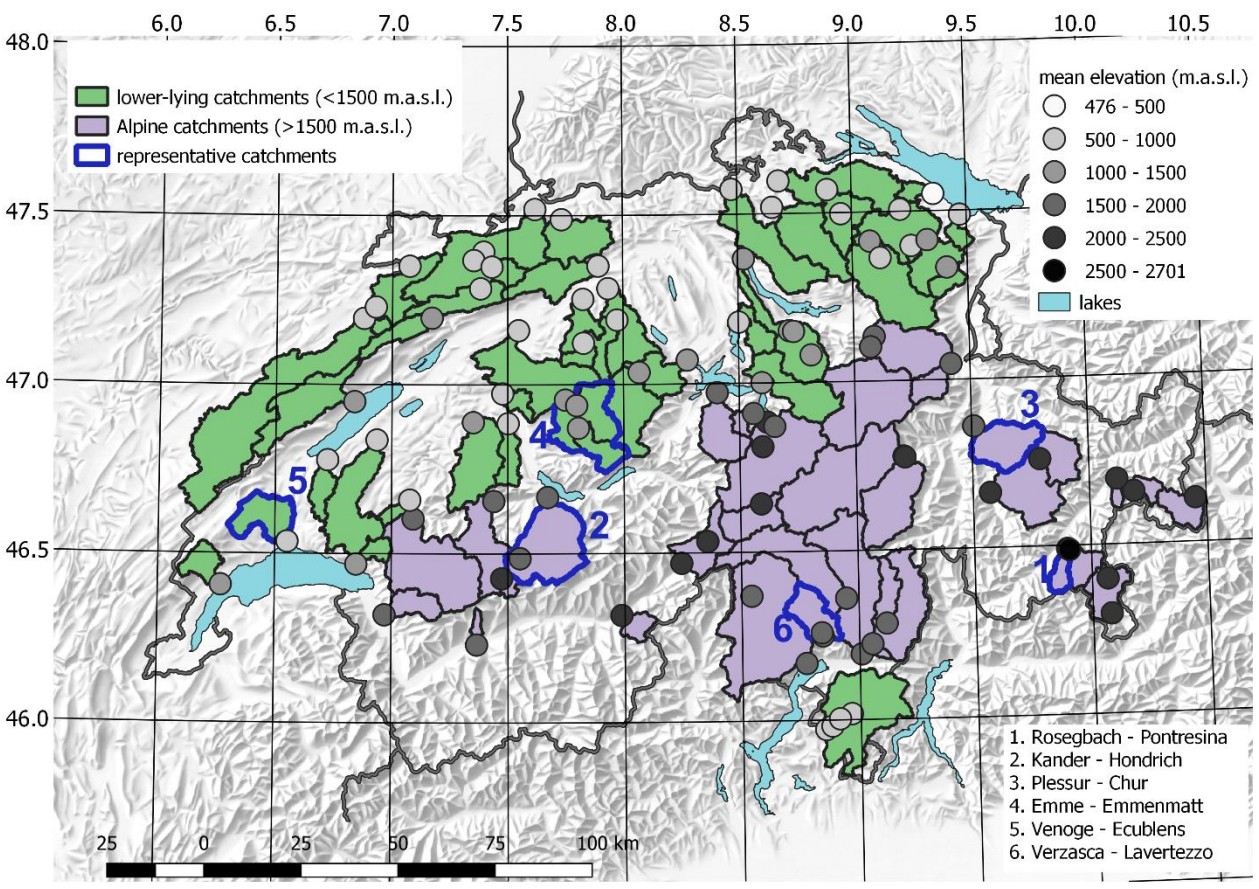

**Figure 1: Overview of the study catchments and the location of the corresponding gauging stations (dots). Grey shadings show the mean elevation of a catchment. Catchments are divided into two groups: lower-lying catchments (green) and Alpine catchments (purple). Blue contours indicate the six example catchments: Rosegbach (1), Kander (2), Plessur (3), Emme (4), Venoge (5), Verzasca (6).**

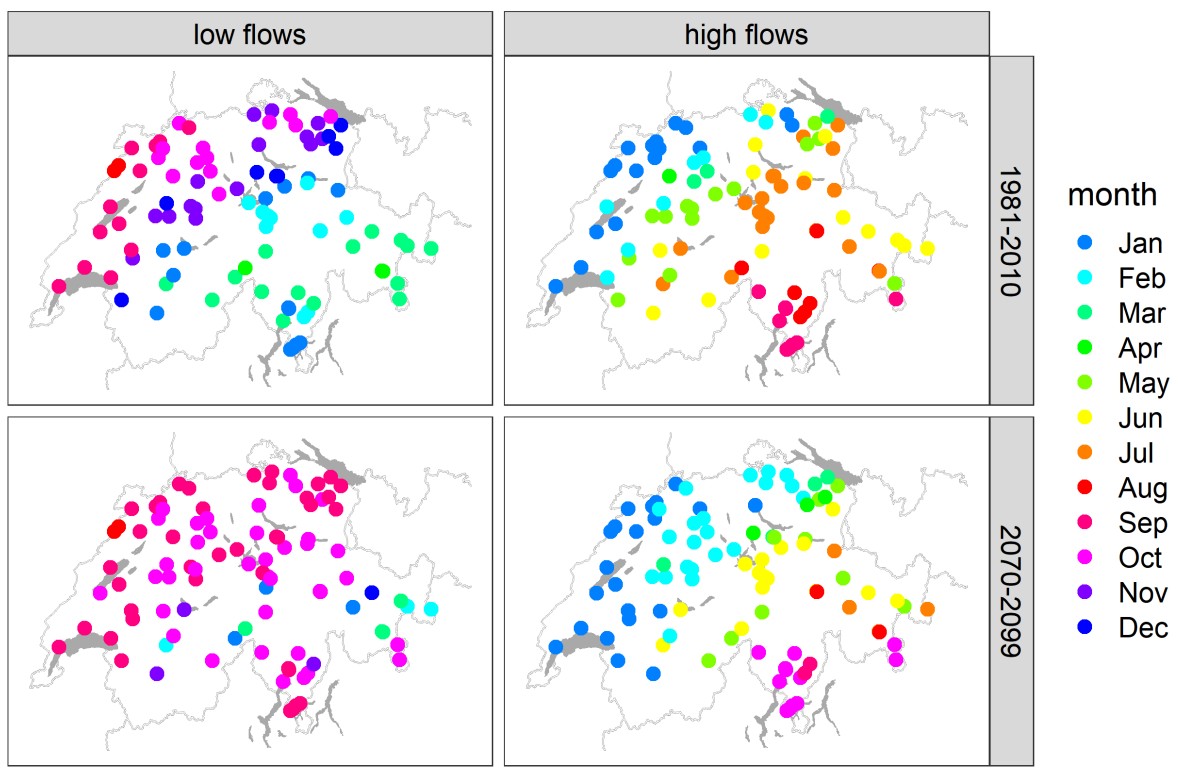

**Figure 2: Median monthly occurrence of moderate low flows (left panels) and high flows (right panels) for the reference period (1981–2010, upper panels) and for the end of the century (2070–2099, lower panels).**

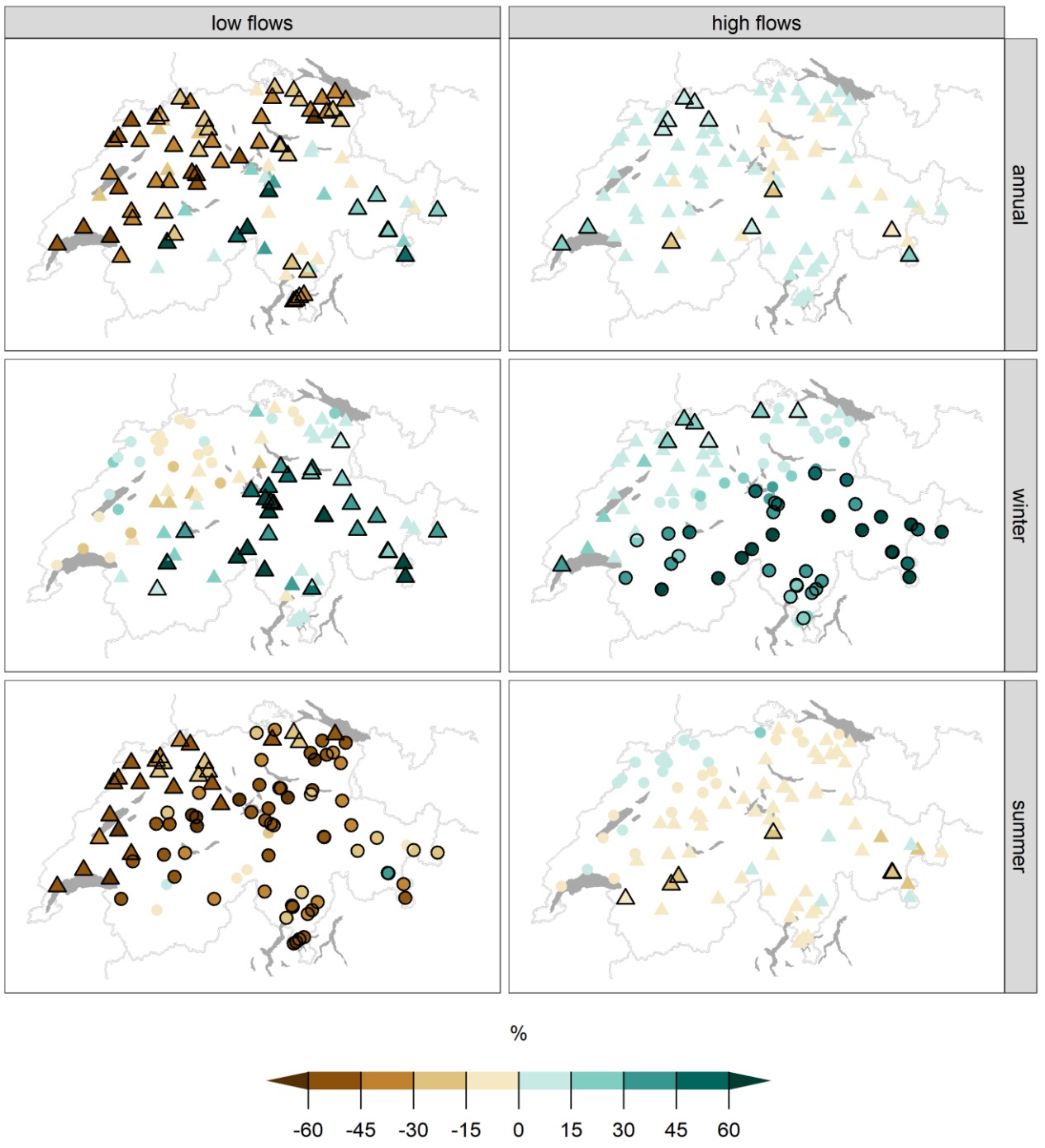

**Figure 3: Relative changes of magnitude by the end of the century for moderate low flows (left panels) and moderate high flows (right panels) for the year (upper panels), the winter half-year (middle panels), and the summer half-year (lower panels). Triangles indicate catchments with annual moderate low and high flows occurring in the time window in the reference period. Circles indicate seasonal low and high flows outside the typical low- and high-flow season. Black contours indicate changes with at least 90% of the models agreeing on the direction of change.**

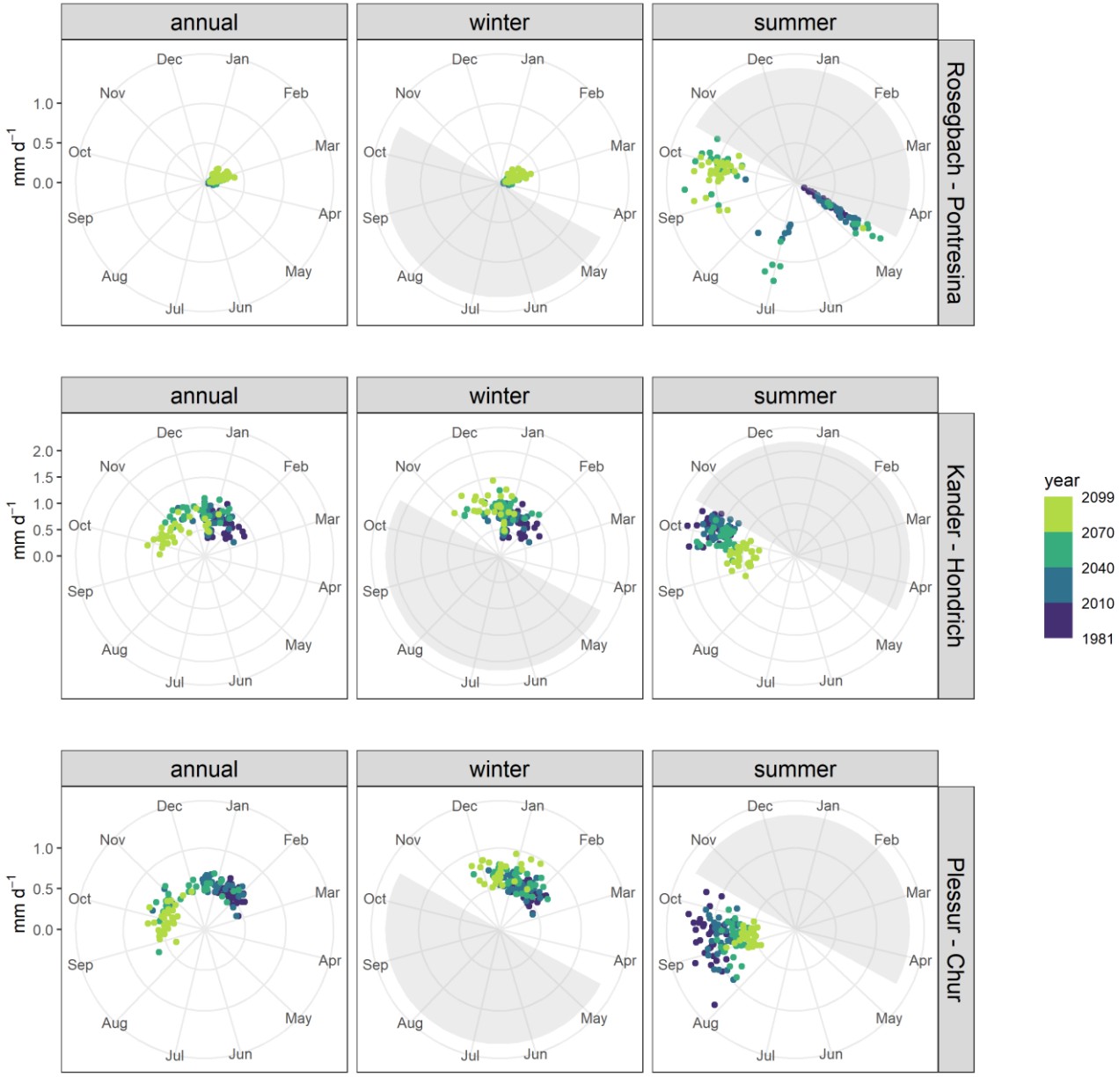

**Figure 4: Multimodel median of magnitude and seasonality of low flows and seasonal low flows in Alpine catchments: Rosegbach (top row), Kander (middle row), and Plessur (bottom row).**

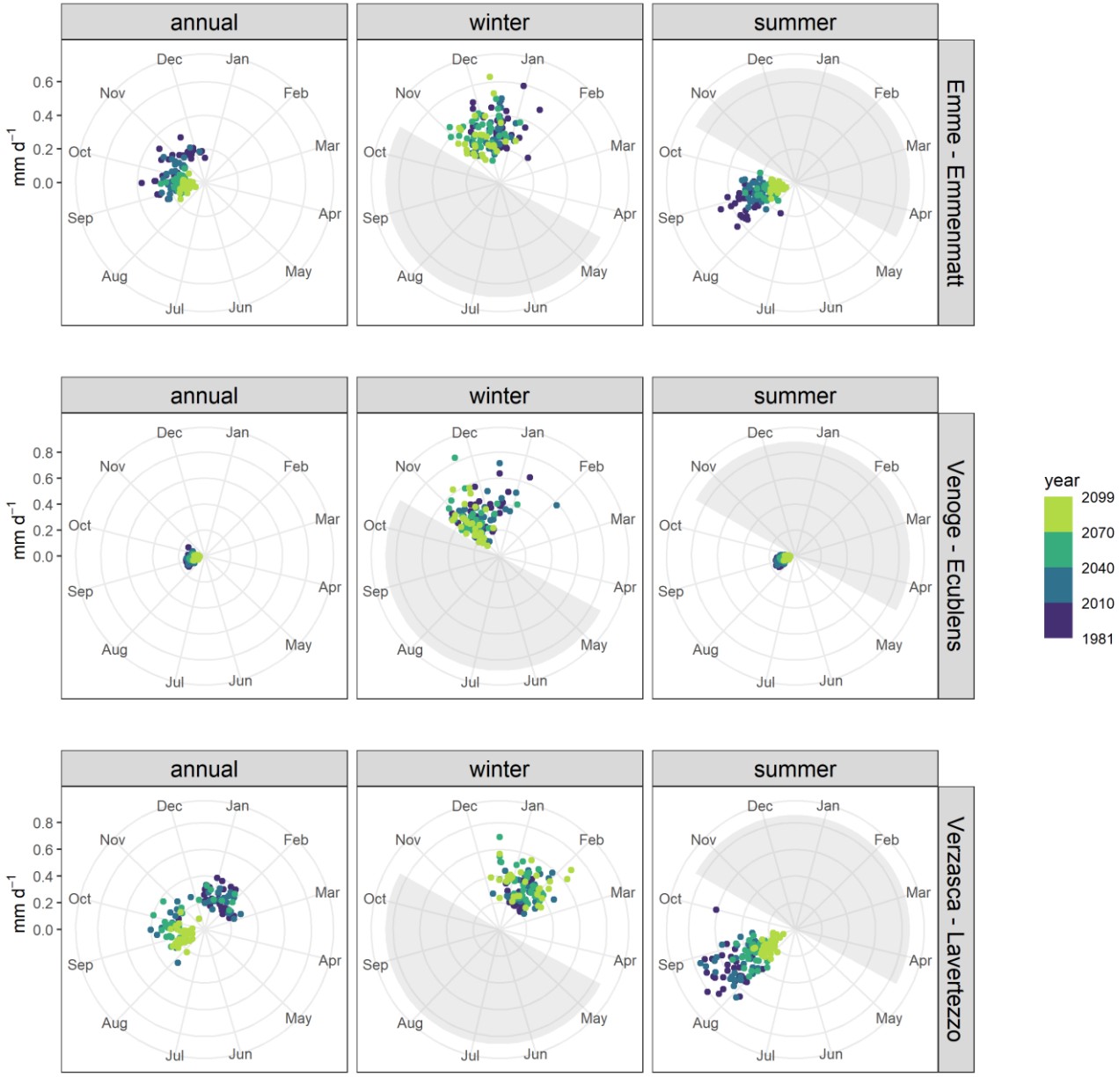

**Figure 5: Multimodel median of magnitude and seasonality of low flows and seasonal low flows in lower-lying catchments: Emme (top row), Venoge (middle row), and Verzasca (bottom row).**

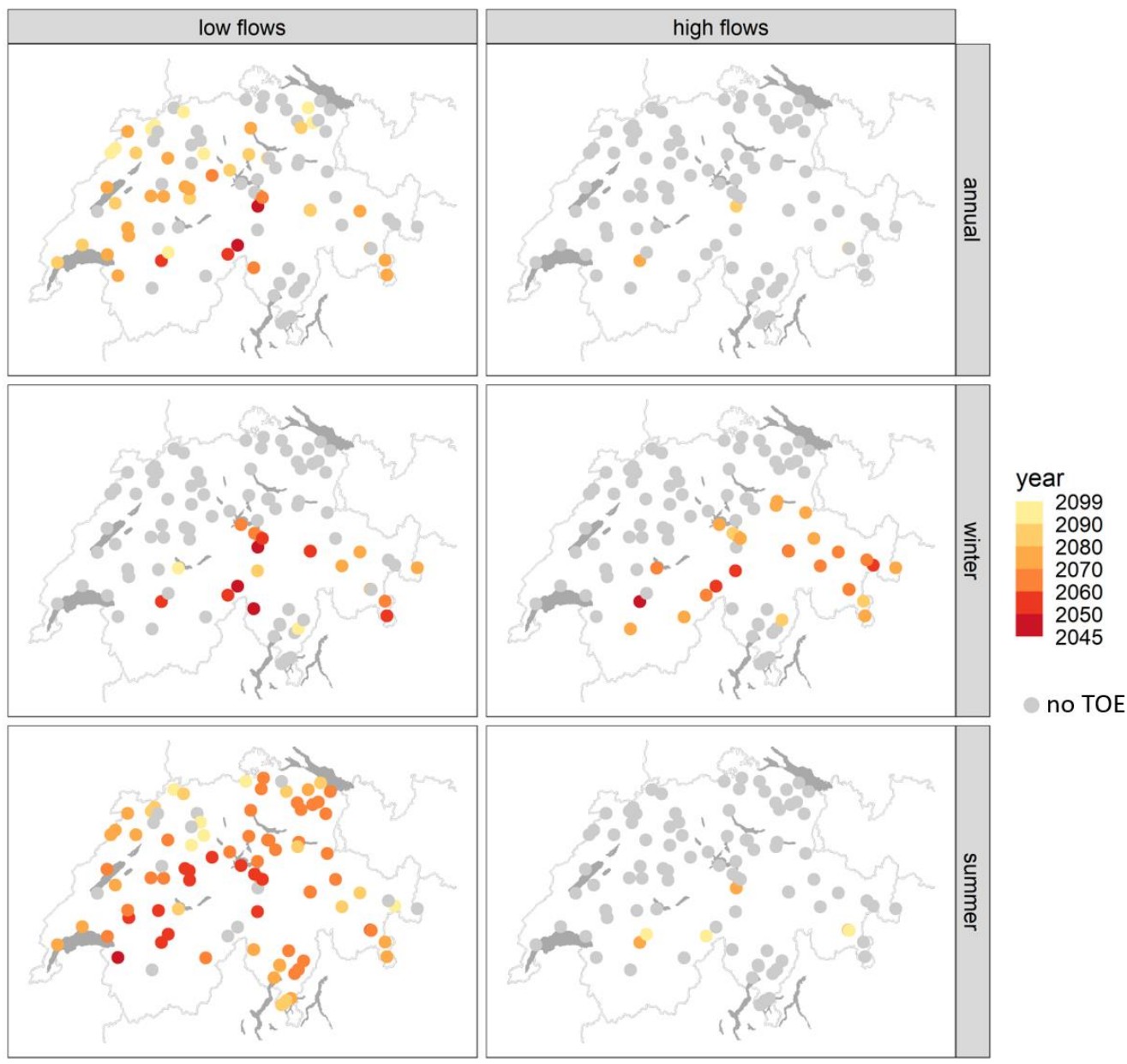

**Figure 6: Time of emergence of moderate low flows (left panels) and moderate high flows (right panels) when at least 66% of the models agree on significant changes in the distribution of moderate low and high flows.**

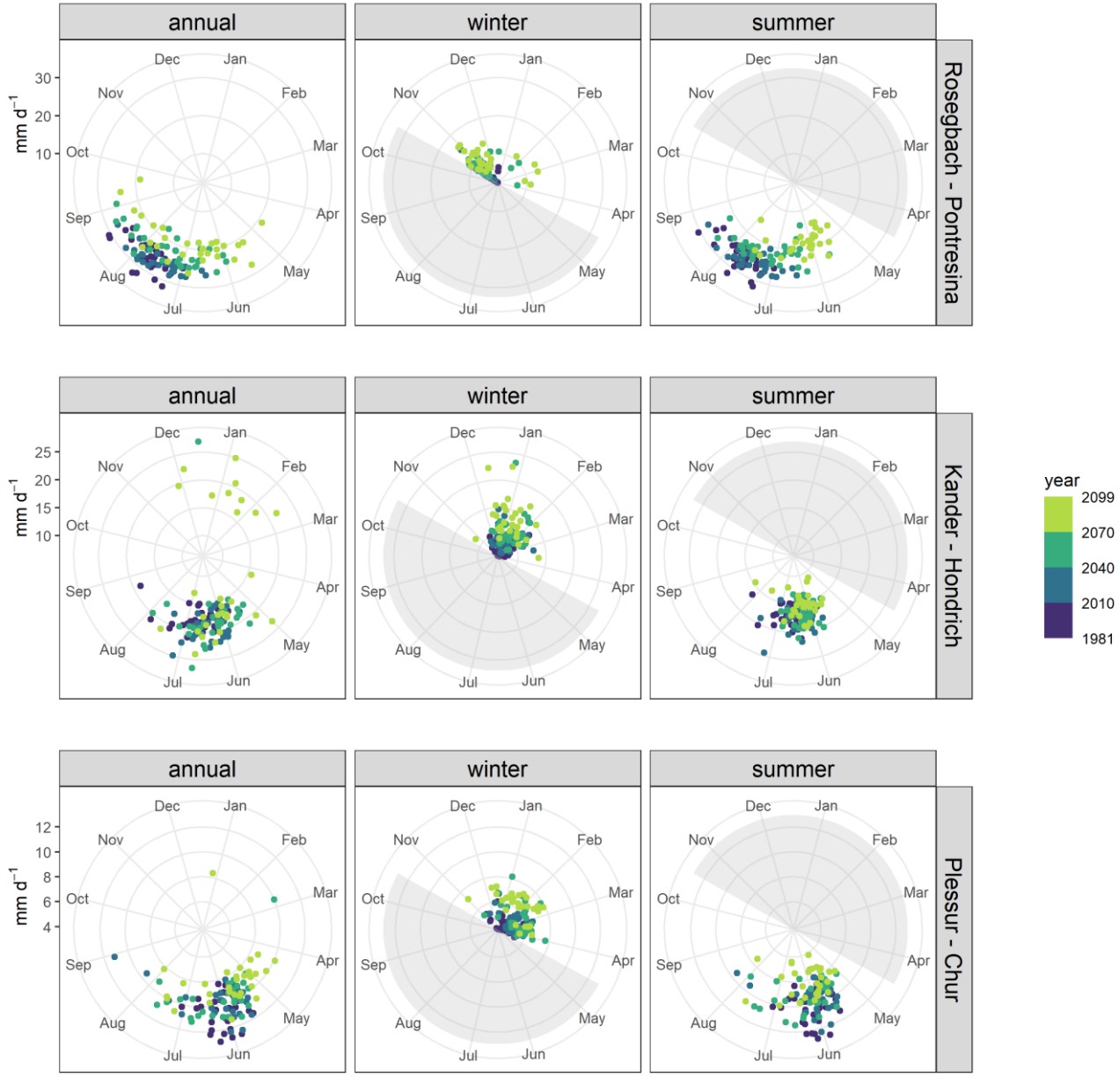

**Figure 7: Multimodel median of magnitude and seasonality of high flows and seasonal high flows in Alpine catchments: Rosegbach (top row), Kander (middle row), and Plessur (bottom row).**

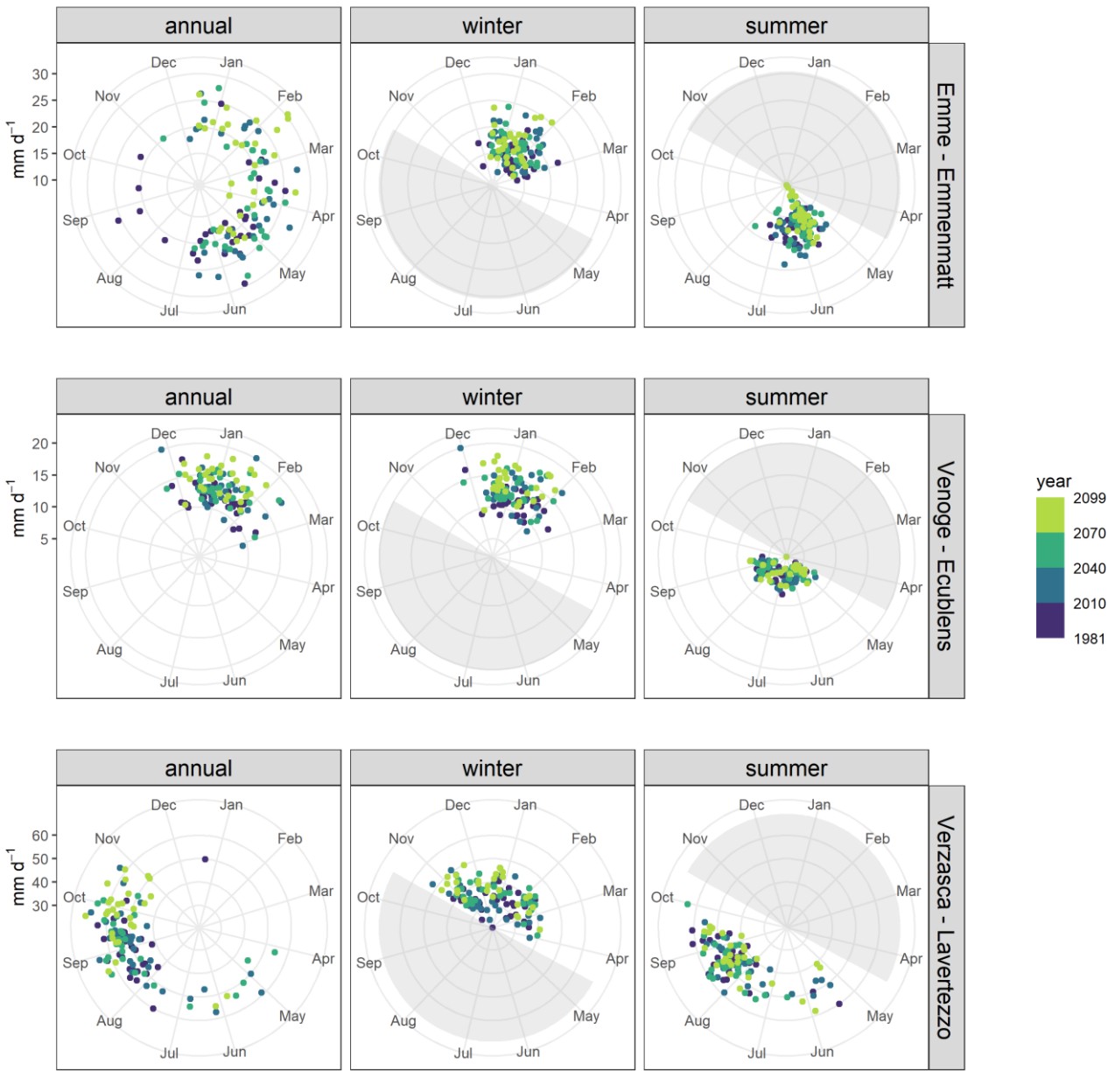

**Figure 8: Multimodel median of magnitude and seasonality of high flows and seasonal high flows in lower-lying catchments: Emme (top row), Venoge (middle row), and Verzasca (bottom row).**

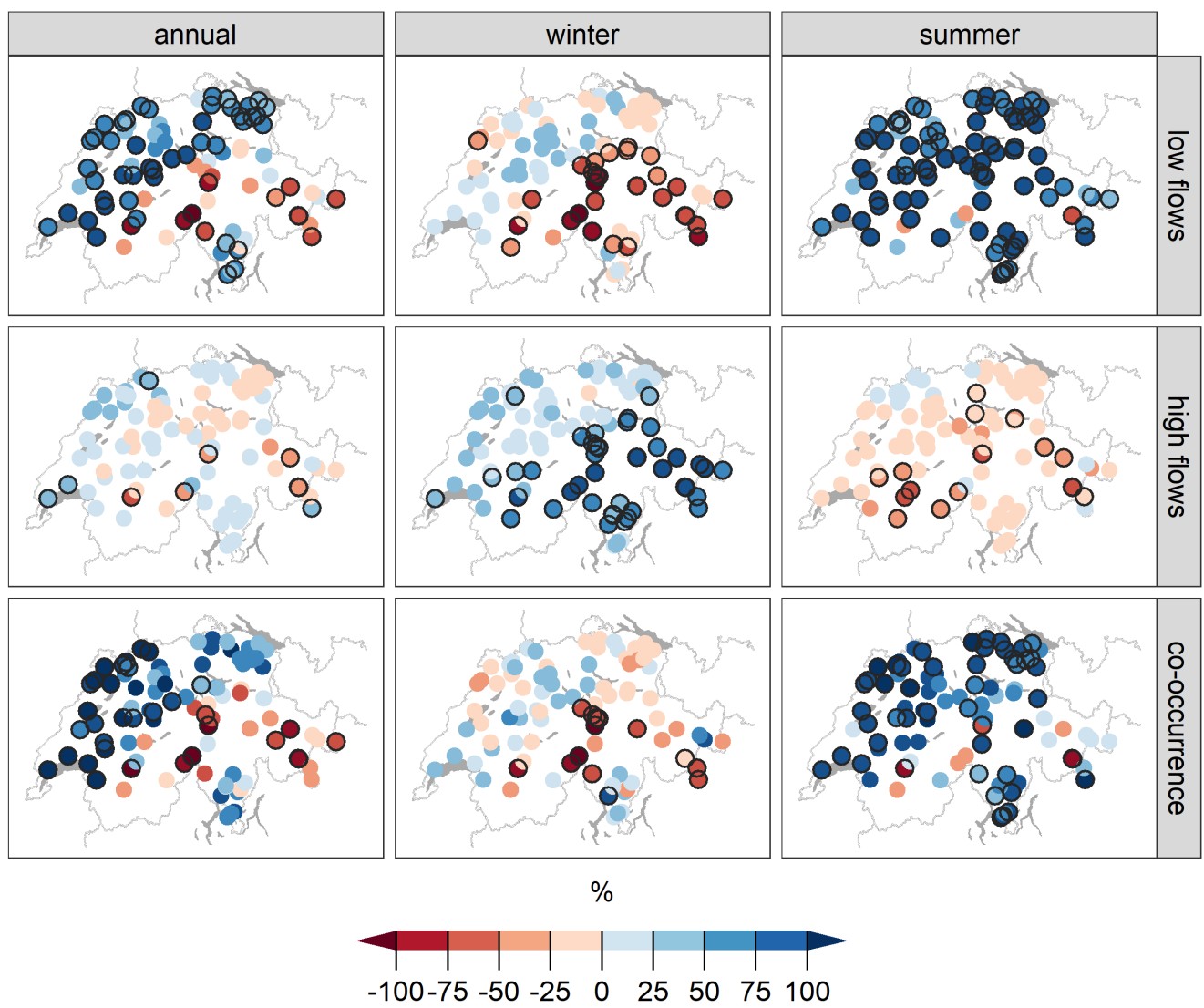

**Figure 9: Relative changes in the frequency of occurrences of moderate low-flow events (lower than the median of the reference period, top panels), of moderate high-flow events (higher than the median of the reference period, middle panels), and co-occurrence of low- and high-flow events (bottom panels) by the end of the century. Black circles indicate changes with at least 90% of the models agreeing on the direction of change.**