# Peer review of "River runoff in Switzerland in a changing climate – changes in moderate extremes and their seasonality"

_Hydrology and Earth System Sciences, 2020_

## Author Response (AR1)

**POINT BY POINT REPLIES AND LIST OF MAJOR CHANGES**

**Referee #1**

The article analyses climate change impacts on runoff regimes in 93 rivers in Switzerland. The study is based – like a companion paper by the same authors (submitted as hess-2020-516) – on the results of a large Swiss research project, which provided consistent downscaled local climate projections under three emission pathways. While the earlier paper presents seasonal and yearly changes in the mean discharge, this contribution analyses the same data (simulated daily runoff for the period 1981-2099) with respect to moderate runoff extremes, both high and low flows. The dataset is available in a publicly available data repository (Zenodo), and is described in a corresponding paper in Geoscience Data Journal, which is under review.

After some careful revisions, which refer mainly to a clearer and more consistent style, the article certainly deserves publication, also because it complements the previous companion paper.

*Reply: Thank you very much for your thorough review. We very much appreciate your inputs and suggestions.*

My review is provided in 2 documents: 1) a PDF document with rather general remarks and recommendations, and 2) detailed PDF annotations directly in the manuscript.

**General remarks**

Summer/winter: there are several "seasonal" analyses, referred to as "extended summer" or "extended winter", which are actually half-years. In some places, just "summer" and "winter" is used. I suggest to use the terms "summer half-year" and "winter half-year" consistently throughout the text to avoid confusion with the common meaning of a season. This is important, because in the context of seasonality shifts, you also refer to the usual seasons, like "late summer" and "early autumn" (e.g. L 24). Please rewrite first paragraph in section 3 accordingly.

*Reply: Thank you for this suggestion. We referred to summer half-year and winter half-year in the text to avoid confusion.*

Low and lowest flows: In the results and discussion sections, there are frequent uses of the terms "low" and "lowest" flows, which most likely refer to the same. If that is true, use "low flow" consistently. Analogously, for the high and highest flows.

*Reply: The intention to use a distinction between low flows and lowest flows (and for high/highest flows accordingly) was based on the seasonality of low and high flows. For example, low flows in summer in Alpine catchments are not absolute low flows but relative low flows. That's why we highlighted the absolute low/high flows with triangles in Figure 2. We tried to make this clear by using "low flows" for absolute low flows and "lowest seasonal flows" for relative low flows in the season. In the revised manuscript we referred to annual low/high flows and low/high flows in the summer/winter half year, respectively.*

"alpine or Alpine": lower case refers, in my opinion, to "mountainous", while upper case Alpine means "within the perimeter of the Alps". Since several of the north-western catchments (Jura) are outside the perimeter of the Alps, a clear distinction of the terms is necessary.

*Reply: We agree with your definition and made this clear to avoid confusion.*

It may help to improve readability, if, e.g. in Fig. 1, you add a classification to the legend which indicates the "lower lying catchments", the alpine and high alpine (Alpine?) catchments. (could be added to the mean altitude classification in the legend).

*Reply: Thank you for this suggestion. This has also been raised by the other reviewer. We adapted figure 1 according to your suggestions.*

**Abstract**

Paragraphs 2 and 3 are difficult to read. L 14 says that annual low flows will increase in Alpine catchments. L 15-16 says that "… annual high flows are projected to slightly increase in most catchments but to decrease in high Alpine catchments." Are the "high Alpine catchments" a clearly different subset of catchments than the "Alpine catchments" in L 14? L 16-19: the discussion on the contradiction between moderate extremes and extremes takes too much space in the abstract, since also in the text it is rather a side note (L 298-300).

*Reply: The abstract was rewritten according to your suggestions and the suggestions raised by the other reviewer. Catchments showing a clear decrease in high flows (more than -5% & strong model agreement) have a mean altitude greater than 1800 m.a.s.l. , with one exception. The sentence referring to the decrease in "high Alpine catchments" was changed accordingly.*

Please consider reorganizing the abstract to discuss all effects of low flows (magnitude, time of emergence, shifts in seasonality) first and then analogously for the high flows.

*Reply: Thank you for your suggestions. The organization of the abstract was changed according to your suggestion.*

**Section 2 – Data**

L 111-115: You should point out that RCP 8.5 is considered a worst-case climate change scenario, and therefore possible changes are most severe. The number of simulations is certainly the least important reason.

*Reply: The reasons for using RCP8.5 are discussed in more detail. Several reasons led to the decision to use RCP8.5. For example, RCP8.5 is the worst-case scenario and changes in low and high flows are stronger than under other scenarios which makes the interpretation easier. Also, the number of simulations available in the Hydro-CH2018-Runoff ensemble differs between the RCPs which may affect some of the results. And last, a large number of simulations within an emission scenario increases the robustness of our results, particularly for the analysis of the time of emergence.*

**Section 3 – Methods**

See first general remark above.

More remarks are given in the manuscript PDF.

*Reply: We copied your remarks from the manuscript and answer them later in this file.*

**Section 4 – Results**

Please check for "Alpine" or "alpine". Are all the lower lying catchments outside of the Alps?

*Reply: We adapted figure 1 to make a distinction between the lower lying catchments and Alpine catchments. All catchments outside the Alpine area have a mean elevation below 1500 m.a.s.l. (also catchments in the Jura mountains have a mean altitude below 1500m.a.s.l.) A clear distinction between "Alpine" and "alpine" was made.*

I agree with the outline discussing low flows first and then the high flows. However, Fig. 2, 3 and 6 present both together, i.e., section 4.2 refers back to parts of a previous figure. Consider putting the sub-figures in a grid and label the columns clearly "low" and "high flows". Fig. 2 might benefit if you also label the rows with "Reference" and "Future". Since in sections 4.2 and 4.3 there is no direct confrontation of "low" and "high" with respect to these figures, it might have advantages to split them, i.e. put 2a and 2c in one Fig. to be discussed in section 4.2, 2b and 2d in a separate one to be discussed in section 4.3. Similarly, figures 3 and 6.

*Reply: Thank you for this suggestion. We added a grid and labelled figures 2, 3, and 6 according to your suggestions to make a clear distinction between low and high flow results. We also inserted a grid in the other figures to be consistent in style.*

**Section 5 – Discussion**

Large parts of this section (at least 5.1) appear to continue the presentation of results rather than a "discussion". Real discussion is the last paragraph of 5.1 (which could be enhanced).

*Reply: More emphasis on the implications of our results on other areas has been given.*

Uncertainties (5.5): I am afraid, that this paragraph is not very helpful, since it is purely qualitative. Without having said anything about uncertainties before, you start " … are larger for moderate high flows than for moderate low flows". I would appreciate a statement why you believe that the message of your results is valid, despite various uncertainties, whose quantification certainly would exceed the scope of this paper.

*Reply: We agree that the quantification of uncertainties would exceed the scope of this paper. But we discussed the associated uncertainties in more detail and added a statement on the validity of our results.*

**Section 6 – Conclusions**

In my opinion, this is more a Summary than Conclusions. In fact, the only clear "conclusion" is from L362-365. I suggest to name the section "summary and conclusions" and add some conclusions also related to the high flows.

*Reply: We renamed the title of this section to "summary and conclusions" and included conclusions related to high flows.*

Evaluation of review criteria (more details are provided by annotations in the PDF manuscript):

1. Does the paper address relevant scientific questions within the scope of HESS? Yes. The focus is on rivers in Switzerland, but methods and results are certainly of interest for other (alpine) countries.

2. Does the paper present novel concepts, ideas, tools, or data? Yes.

3. Are substantial conclusions reached? Yes.

4. Are the scientific methods and assumptions valid and clearly outlined? Yes.

5. Are the results sufficient to support the interpretations and conclusions? Yes.

6. Is the description of experiments and calculations sufficiently complete and precise to allow their reproduction by fellow scientists (traceability of results)? All directly used datasets are publicly accessible. Therefore, the reproduction of the specific results in this paper should be possible.

7. Do the authors give proper credit to related work and clearly indicate their own new/original contribution? Yes.

8. Does the title clearly reflect the contents of the paper? Yes.

9. Does the abstract provide a concise and complete summary? See comments above.

10. Is the overall presentation well structured and clear? In general, yes.

11. Is the language fluent and precise? As in the previous paper, in my opinion, many paragraphs seem to be too closely translated from German, which makes the text difficult to read.

*Reply: The revised manuscript has been edited by a professional English editor.*

12. Are mathematical formulae, symbols, abbreviations, and units correctly defined and used? N.a.

13. Should any parts of the paper (text, formulae, figures, tables) be clarified, reduced, combined, or eliminated? See comprehensive PDF annotations, both to text and figures, directly in the manuscript

14. Are the number and quality of references appropriate? Yes.

15. Is the amount and quality of supplementary material appropriate? Dataset available at Zenodo.

**Detailed PDF annotations directly in the manuscript.**

*Reply: Thank you for the annotations in the manuscript. We copied the comments which need some more discussion and answer them point by point below.*

l16: To avoid confusion, I suggest to use "summer half-year" and "winter half-year" consistently throughout the text.

*Reply: The terms "summer half-year" and "winter half-year" are used.*

L37: A streamflow drought is defined as a period when the discharge is below a given threshold level. A distinction is made between the minimum low flow discharge of a drought event, and the corresponding deficit volume and duration of the event.

Did you really mean "streamflow drought" or rather "minimum low flow discharge ..."?

*Reply: Thank you for the explanation. We referred to "extreme low flows" in line 37.*

L56: "Central Europe" is not a proper opposite to "Alpine" - the Alps are certainly part of Central Europe.

*Reply: We fully agree with your point and rephrased this sentence.*

L60: I don`t understand the meaning of this phrase.

*Reply: We removed the sentence "Past events can help to put potential future changes into perspective."*

L100: It is certainly not a specific issue in this paper, but I have concerns about this calibration/validation strategy. I learned to use an independent time period for validation. If consecutive years are used for calibration and validation, then, e.g. a possible trend of the model's output or system state is compensated by this choice of interwoven calibration/validation periods.

*Reply: The calibration and validation of the hydrological model is described in detail in Muelchi et al. (2021): "The intention of using every second year within 30 years for calibration is to minimize the potential effect of random and non-random trends by using too short calibration periods. Climate change is already observed in the period 1985–2014. PREVAH is therefore trained to also simulate runoff under changing conditions. During the calibration process, PREVAH was run for the whole period while comparing the simulations with observations only for even years. We intentionally chose uneven years for the validation period since some of the years include periods of extreme weather such as very dry summer (e.g. 2003), severe floods (e.g. 2005, 2011) or winters with extreme snowfall and thus extreme snowmelt in spring (e.g. 1999). This leads to the assumption that if the model performs well in uneven years (validation), the calibrated parameters produce stable results also for more extreme or changing conditions. "*

*Muelchi, R., Rössler, O., Schwanbeck, J., Weingartner, R., & Martius, O. (2021). An ensemble of daily simulated runoff data (1981–2099) under climate change conditions for 93 catchments in Switzerland (Hydro-CH2018-Runoff ensemble). Geoscience Data Journal.*

And - I think that the correct opposites are "even" and "odd"!

*Reply: The term "odd" was used instead of "uneven".*

L112: The most important reason, in my opinion, should be that RCP8.5 is considered the worst-case scenario.

*Reply: This is indeed one of the most important reasons. We refer to the answer under "data" above.*

L126: This is inconsistent. As I understand, MAM7 uses the same rules, whether it is for a year or a half-year. Therefore, you should not use the superlative "highest" or "lowest" for the half-year values, if the yearly values are called "moderate high (or low)"

*Reply: We refer to the answer above. The intention was to make a distinction between absolute and relative low/high flows.*

L129: Is it obvious, that this the median of 20 means?

*Reply: We added an explanation here. "The multi-model median of the relative changes by end of the century in 20 simulations is regarded as the best estimate. "*

L141: This needs clarification similar to the previous companion paper.

*Reply: This part has been rewritten similar to the revised version of the companion paper. The concept of time of emergence is based on a statistical test between two distributions (reference vs future). However, changes in low/high flows may not be linear over time. Even though changes in low/high flows are tested as significant in one period, they may not be significant in all periods afterwards (e.g., due to non-linear effects of enhanced snow melt, decreasing snow cover, increasing glacier melt and decreasing glaciation).*

L218: This is the first time that you mention the Jura. So far, the catchments in that region seemed to be just "the lower-lying catchments"? Or do I miss something?

*Reply: We removed the term "Jura" and only mentioned "catchments in north-western Switzerland".*

L234: This section title is not clearly describing its content. What about: Changes in frequency and co-occurence of low and high flow events? (as in L235-236)

*Reply: We changeed the section title according to your suggestion, thank you.*

L237: Do you mean a value with a return period of 2 years? "occuring every second year" is certainly not correct

*Reply: We agree with your point. This sentence was changed to "The threshold runoff value is defined as the median in the reference period."*

L252: I would move this definition to the begin of the section

*Reply: The definition was moved to the beginning of the section.*

L271: what is "very high"? Since the study refers to 93 Swiss catchments, it should be possible to give a threshold altitude

*Reply: A threshold altitude (>2300 m.a.s.l.) has been given.*

L316: is "robust" the correct word for this?

*Reply: We removed this sentence in the revision process.*

Figure 1: It would help to add to the altitude legend a classification that is used in the text, like "low lying ..", "high alpine ...", etc.

Also, "Jura" and "Alps" should be appropriately labelled.

*Reply: Figure 1 was changed according to your suggestions and the suggestions raised by the other reviewer.*

Figures 2/3: You may consider to organise the figures as a grid and label columns clearly "low" and "high", rows "reference" and "future".

Or maybe split in 2 Figures, one for low, and one fior high.

*Reply: Thank you for this suggestion. We added a grid and label the figures 2, 3, and 6 according to your suggestions to make a clear distinction between low and high flow results.*

Figure 4: The scales at the left of each fig must be labeled! I assume, it is intensity, but don't know for sure what this really means. Is it discharge, or specific discharge?

Labeling of the color bar "year" should start with 1980 and end with 2100.

Applies also to figs 5, 7, 8!

*Reply:* *Thank you. Figures 4, 5, 7, 8 were changed according to your suggestions.*

**Referee #2**

The authors have conducted a study over 93 Swiss catchments that cover a range of elevations. They analyzed moderate annual and seasonal low and high flows under the influence of climate change (RCP 8.5), using a modeling chain comprising of 20 RCMs and one hydrological model (PREVAH). The high and low flows were evaluated based on (1) magnitude, (2) emergence, (3) changes in seasonality, and (4) frequency.

With careful revisions, I believe the article will be ready for publication. My comments below are organized by the respective section of the paper they are relevant for.

*Reply: Thank you very much for your thorough review. We very much appreciate your inputs and suggestions.*

**Abstract:**

From the text, it can be gleaned that the annual and seasonal moderate high and low flows were analyzed according to: (1) magnitude, (2) emergence, (3) changes in seasonality, and (4) frequency. However, this is not explicitly stated in the abstract – it is not until lines 79-81 within the introduction that the analysis is clearly laid out. I would suggest that the authors spend some time reworking the abstract so that the reader can very quickly understand what was analyzed.

*Reply: The abstract was rewritten according to your suggestions and the suggestions raised by the other reviewer. Also, we made clear that we analyze low and high flows in terms of magnitude, emergence, seasonality, and frequency.*

Line 13 – suggest removing the term 'downscaled' before regional climate models

*Reply: We removed this term.*

It would be nice to highlight the changing behavior of high flows per season (see lines 16 and 28/29, where high flows are referred to), as was done for low flows (see lines 15 and 26/27). If the climate model agreement is too poor to draw conclusions on changes to high flows per season, I think this needs to be explicitly stated in the abstract.

*Reply: We discussed changes in high flows in more detail in the abstract and mentioned the difficulties to draw conclusions because of the weak model agreement.*

**Introduction:**

Lines 58-59 read, "high flows may also cause severe damages and significant costs. Hence, potential changes in high flows have to be integrated in water management and infrastructure planning, as well". It seems much more common that infrastructure planning is made resilient against very extreme events. Also, severe damages from flooding are more so associated with more extreme events rather than moderate extremes as defined here. I was hoping that the authors can either support their language more clearly and state how these examples are relevant for moderate extremes, otherwise I would suggest that the language be toned down.

*Reply: Thank you for pointing this out. We rephrased this sentence toning down the language and added that moderate extremes are also important for water use and ecology.*

**Data:**

On line 89, the authors mention that 22 glaciated catchments were analyzed, but the number of catchments representing the other regime types considered are not provided. Suggest adding these numbers to make the text clearer.

*Reply: Numbers were added and Figure 1 was changed to make a clear distinction between high-elevation and low-elevation catchments.*

At least one sentence should be dedicated to why RCP8.5 was selected as the sole scenario, as opposed to the others available, and an explanation of what that pathway represents.

*Reply: The reasons for using RCP8.5 was discussed in more detail. Several reasons led to the decision to use RCP8.5. For example, RCP8.5 is the worst-case scenario and changes in low and high flows are stronger than under other scenarios which makes the interpretation easier. Also, the number of simulations available in the Hydro-CH2018-Runoff ensemble differs between the RCPs which may affect some of the results. And last, a large number of simulations within an emission scenario increases the robustness of our results, particularly for the analysis of the time of emergence.*

Figure 1- please indicate blue shading is for water bodies. Also, see my first comment within the Results section below.

*Reply: Figure 1 was changed according to your suggestions and the suggestions raised by the other reviewer.*

**Methods:**

The authors state that the time of emergence 'may not be stable' (line 141). Could the authors expand upon what they mean by 'stable'? Did the authors also find when the KS test is rejected repetitively? This indicator could seemingly be made more robust by requiring more than just one rejected KS test.

*Reply: The concept of time of emergence is based on a statistical test between two distributions (reference vs future). However, changes in runoff may not be linear over time. Even though changes in low/high flows are tested as significant in one period, they may not be significant in all periods afterwards (e.g., due to non-linear effects of enhanced snow melt, decreasing snow cover, increasing glacier melt and decreasing glaciation). We included two figures in the Supplement showing the rejections of the p-value over time for low and high flows (see figure 1 below). Further explanation on this point were made in the methods section.*

[Figure]

*Figure 1: Temporal evolution of rejections of p-value for low flows.*

The model agreement that you are highlighting for magnitude (Figure 3) and frequency (Figure 9) is 90%, whereas the model agreement highlighted for emergence (Figure 6) is 66%. Can you please explain your reasoning for highlighting different levels of model agreement?

*Reply: We chose these thresholds following the IPCC terminology where 90% refers to "very likely" and 66% to "likely". The threshold of 66% is used for the determination of time of emergence. The definition of time of emergence in this study relies on the definition used by Mahlstein et al. (2011). We extended the description of time of emergence and added a statement on the threshold definitions.*

*Mahlstein, I., Knutti, R., Solomon, S., and Portmann, R. W.: Early onset of significant local warming in low latitude countries, Environ. Res. Lett., 6, 3, 034009, 10.1088/1748-9326/6/3/034009, 2011.*

**Results:**

The results are described in terms of low and high elevation regions, however the reader is not provided a clear picture or threshold of how the authors have separated these grouped catchments nor are these regions indicated within Figure 1 (please see for instance Figure 2 from Brunner et al., 2019 – Science of the Total Environment: https://www.sciencedirect.com/science/article/pii/S0048969719306576).

*Reply: Figure 1 was changed according to your suggestions and suggestions raised by the other reviewer.*

For Figures 3, 4, 5, 6, 7, 8, 9 – would suggest replacing the word 'YEAR' with 'ANNUAL' since this corresponds to the terminology used in the main body text.

*Reply: Thank you for highlighting this inconsistency. "YEAR" was replaced with "ANNUAL".*

Figures 3 and 9 indicate where models agree 90% of the time, whereas Figure 6 indicates when 66% of the models agree. Also, Figure 6 uses grey circles to show non-agreement, which is different from Figures 3 and 9, which use muted tones. What is the reason for designing these figures so differently? Where possible, suggest making the figure design cohesive.¨

*Reply: The content of figures 3 and 9 requires a different visualization than figure 6. Figures 3 and 9 indicate changes over time. There is a value (%) for the multimodel median for every catchment. However, not all changes are robust among the climate models. Therefore, we highlight catchments with a strong model agreement (>=90% of models) on the sign (positive or negative) of change with a black circle. Figure 6 shows the time of emergence and such a time of emergence cannot be determined for all catchments. For such cases we just use a grey color. This has been added to the legend.*

**Discussion:**

On lines 287-290 and 315-317 you offer some discussion of the significance of your work, but this comes across as very brief. In general, the discussion section is a good opportunity to substantiate the overall implications of this research. Indeed, this work is relevant for the agricultural and hydropower industries as you mention – I encourage you to make stronger statements about the relevance of your work and look for connections in literature. As an example, I highlight the relevance of your work to Switzerland's hydropower concessions, which are strongly influenced by projected low and high flow behaviour – described here:

https://hess.copernicus.org/articles/24/3815/2020/ and especially by the following authors (just an example of suggested authors):

Dr. Ludovic Guadard

Prof. Dr. Fanco Romerio

Prof. Dr. Hannes Weigt

*Reply: Thank you very much for this comment. The discussion section will was extended by statements on the significance of our study on other areas. Thank you also for the reference and the list of authors.*

On line 292: '…will decrease in the projections' should be changed to '…has been shown to decrease within the projections' or '…is likely to decrease in the future'.

*Reply: The sentence was changed according to your suggestion.*

**Conclusion:**

Any mention of the greater implications of your work is generally absent from your conclusion section (please see similar comment in the discussion section above).

*Reply: See answer above. A statement on the implications of our study was made.*

**List of most relevant changes to the manuscript:**

**General:**

- Rephrasing of unclear sentences
- Corrections according to suggestions by English editor
- Changed "lowest seasonal flows" to low flows in the summer/winter half-year
- Changed "highest seasonal flows" to high flows in the summer/winter half-year
-

**Title:**

- Changed to "River runoff in Switzerland in a changing climate – changes in moderate extremes and their seasonality"

**Abstract:**

- Rearranged the order of our statement
- Shortened

**Introduction:**

- Highlighting differences between new hydrological scenarios and previous scenarios

**Data:**

- Add more information on reasons to choose RCP8.5

**Methods:**

- Time of emergence: mostly rewritten to make it clearer and to explain the method behind time of emergence; choice of thresholds and methods explained; explanations on temporal evolution of time of emergence added

**Results:**

- Add information on persistence of time of emergence

**Discussion:**

- Examples of potential implications on various sectors discussed in more detail
- Uncertainties and limitations further discussed

**Figures:**

- General: Figures 2-9 were put into grids and labelled
- Fig. 1: added classification between lower-lying catchments and Alpine catchments
- Figs S1 and S2: new figures for the rejection of p-values over time showing the persistence of time of emergence